# A Pruning Method Based on Feature Map Similarity Score

**Jihua Cui [1], Zhenbang Wang [2], Ziheng Yang [3,\*] and Xin Guan [4]**

1    Department of Power Engineering, Harbin Electric Power Vocational and Technology College,
     Harbin 150030, China; lhf_cjh@sina.com
2    Dispatching and Control Center, State Grid Heilongjiang Electric Power Company, Ltd.,
     Harbin 150090, China; zhenbangw@163.com
3    School of Electronic Engineering, Heilongjiang University, Harbin 150080, China
4    School of Data Science and Technology, Heilongjiang University, Harbin 150080, China; gx.hlj@outlook.com
*    Correspondence: yzh@hlju.edu.cn

**Abstract:** As the number of layers of deep learning models increases, the number of parameters and computation increases, making it difficult to deploy on edge devices. Pruning has the potential to significantly reduce the number of parameters and computations in a deep learning model. Existing pruning methods frequently require a specific distribution of network parameters to achieve good results when measuring filter importance. As a result, a feature map similarity score-based pruning method is proposed. We calculate the similarity score of each feature map to measure the importance of the filter and guide filter pruning using the similarity between the filter output feature maps to measure the redundancy of the corresponding filter. Pruning experiments on ResNet-56 and ResNet-110 networks on Cifar-10 datasets can compress the model by more than 70% while maintaining a higher compression ratio and accuracy than traditional methods.

**Keywords:** network pruning; redundant filters; similarity; ResNet

## 1. Introduction

Artificial intelligence based on deep learning is one of the most promising and full-of-potential technologies today. It can use big data and powerful computing power to automatically learn from data, extract useful features, and make predictions and decisions through models to realize various intelligent applications. Artificial intelligence based on deep learning can play a role in healthcare, transportation, finance, energy, education, and other fields to achieve more accurate and efficient decision-making and prediction and promote the progress of science and technology. For example, in the medical field, artificial intelligence based on deep learning can improve the accuracy and efficiency of medical diagnosis by analyzing a large amount of medical data and helping doctors formulate more reasonable and personalized treatment plans. In terms of daily life, intelligent analysis of human health status and human–machine interaction can be carried out through wearable devices and information theory [1–4]. In the field of transportation, artificial intelligence based on deep learning can optimize traffic flow and improve traffic safety and efficiency by analyzing traffic data.

At this stage, deep learning networks such as AlexNet, GoogleNet, VGG-16, ResNet-152, etc., have hundreds of model layers and have continuously achieved excellent results in various tasks such as recognition and classification. However, as the depth of the deep learning network model increases, the number of parameters and calculations increases dramatically. For example, when ResNet-152 performs forward inference on $224 \times 224$-sized images, the number of model parameters reaches 138 million, the amount of floating-point operations reaches 15.5 billion, and the storage footprint reaches 528 MB. This puts forward higher requirements for model deployment. In some scenarios wherein resources, power consumption, and computing power are limited, it is difficult to deploy algorithms into applications [5,6].

Therefore, it is necessary to study how to compress the model in order to improve its efficiency. The researchers found that, although the number of depth model parameters at this stage is large, there is a large amount of redundancy [7]. If you remove some parameters, you can make the model smaller while keeping its performance from being greatly affected. This makes it easy to deploy models on resource-constrained devices. The current research mainly focuses on three aspects: model pruning, knowledge distillation, and model quantification; it also designs new lightweight networks from the model itself. This paper mainly studies the solution to the model's pruning direction.

In this paper, the problem that the heuristic filter importance metric may not correctly reflect the importance of the filter and the problem that the cluster-based pruning method selects the number of cluster centers are complex. A similar pruning scheme is proposed, and the similarity of each feature mapped to all other feature maps is calculated according to the feature mapping diagram output by the filter. The similarity score for each feature map is then calculated to measure the similarity of each filter, removing those with low similarity. This method combines the output of the filter and uses feature map similarity to guide the removal of redundant features.

## 2. Related Works

The general pruning process is to pre-train a network model, then measure the importance of the parameters in the model, remove some parameters according to their importance, and finally fine-tune the network to restore its accuracy. Therefore, a key research aspect of pruning is how to measure the importance of parameters or filters. According to the different pruning objects, it is divided into structured pruning and unstructured pruning.

### 2.1. Unstructured Pruning

Unstructured pruning removes some of the neuronal parameters in the filter, which is a more fine-grained pruning. The importance of a weight is measured primarily by the absolute value of the parameter. Han et al. [8,9] use the L1 norm and the L2 norm to measure the importance of weight values and believe that weight values with small norm values have little effect on the results of the network. Han compresses the model by more than 10 times, and the network performance is basically undegraded. Guo et al. [10] improved Han's method by dynamically connecting pruning in the fine-tuning stage, resetting the value of mask, and reducing the important weight of erroneous pruning. Finally, pruning on LeNet-5 and AlexNet can compress the number of parameters to 108 and 17.7 times, respectively, without loss of network performance. N. Lee and M. Alizadeh perform one-shot pruning before the network model is initialized [11,12]. Based on the importance of the weight connection, determine the importance of the weight and remove the weight with low importance. It can save time spent on fine-tuning.

### 2.2. Structured Pruning

Unstructured pruning results in sparse filters that do not allow for acceleration on general-purpose hardware. Although the number of parameters is reduced, they cannot be effectively accelerated [13]. Structured pruning, on the other hand, is the removal of a filter or some channels of the convolutional layer, or even the entire convolutional layer. Although the number of parameters removed is large, the impact on the accuracy of the model is still small. And it is convenient for hardware acceleration implementation. Therefore, structured pruning is more popular with researchers. Li et al.'s [14] pruning method is similar to Han's, using the sum of the weight values of the entire filter to measure the importance of a filter and setting a threshold. If the sum of all the weights of the filter falls below the threshold, the filter is considered redundant and removed. He et al. [15] modeled channel pruning as an optimization problem. Channel selection is performed by LASSO regression and least squares to remove unimportant filters. Liu et al. [16] used the scaling factor of the BN layer to measure the importance of the corresponding channel.

If a certain amplification factor tends to 0, then the channel is considered unimportant. Luo et al. [17] propose the ThiNet method to guide filter pruning based on the minimum reconstruction error, and model filter selection as an optimization problem. This method guides the pruning of the current layer network based on the statistics output of the next layer. He et al. [18] proposed the FPGM method, arguing that the filter near the geometic median can be represented by other filters and that the importance of the filter is measured relatively. Lin et al. [19] proposed an HRank pruning method. The pruning method mainly relies on the feature map output of the convolutional layer, and the filter of the convolutional layer is removed with the rank guidance of the feature map. Sui et al. [20] propose the CHIP method. The core idea is to use the independence of the feature map to measure its importance to the response filter. The importance of the channel is measured using the difference in the kernel norm of the feature map matrix before and after a feature map is removed. If the independence of a feature map is low, the feature map is considered dependent on other feature maps and can be removed. Zhuo and Wang prune based on clustering [21,22]. The filter or the feature map of the filter output is clustered, and the filter or feature map that is aggregated into a cluster is considered to be related, so some filters or feature maps in the same cluster can be removed. The summary of related works is shown in Table 1.

**Table 1.** Summary of related works.

| Article | Structure | Criterion | Method |
|---|---|---|---|
| [8,9] | weights | weights magnitude | train, prune and fine-tune |
| [10] | weights | weights magnitude | mask learning |
| [11] | weights | weights magnitude | prune and train |
| [12] | weights | weights magnitude | prune and train |
| [14] | filters | L1 norm | train, prune and fine-tune |
| [15] | filters | filters magnitude | group-LASSO regularization |
| [16] | filters | magnitude of batchnorm parameters | train, prune and fine-tune |
| [17] | filters | output of the next layer | train, prune and fine-tune |
| [18] | filters | geometric median of common information in filters | train, prune and fine-tune |
| [19] | filters | average rank of feature map | train, prune and fine-tune |
| [20] | filters | channel independence | train, prune and fine-tune |
| [21,22] | filters | $L_p$ norm | train, prune, and fine-tune |

Traditional pruning algorithms use zero-value pruning when measuring the importance of the filter, and heuristic metrics are used to measure the importance of the filter. Such methods often require a certain distribution of network parameters to achieve good results. There is another type pruning method based on similarity analysis, considering the similarity between the output feature maps of convolutional layer. This type of method clusters the features output from the convolutional layer, preserves representative feature maps, and then removes filters and channels corresponding to redundant feature maps. However, this type of method often fails to select an appropriate number of cluster centers. Based on existing research, a pruning method based on feature map similarity score is proposed in this paper.

## 3. Method

### 3.1. Algorithmic Thinking

In this paper, the proposed pruning algorithm is based on similarity calculation between the feature maps of each filter output. The idea of the algorithm is that the feature map output by the filter can reflect the feature extraction effect of the filter, and if some feature maps are similar, it means that the feature extraction effect of these filters is similar, and these filters are redundant. Figure 1 is the visualization of the feature map of the first convolutional layer of the ResNet-50 network, and it can be intuitively seen from the figure that there is similarity between the feature maps output by each filter of the

convolutional layer. The two feature maps marked by the red box in the figure have very similar visual performances in terms of light and shade, texture, etc. This means that the filter that outputs these two feature maps extracts similar features such as light and dark, texture, etc. If multiple filters all extract the same or similar features, then these filters are redundant in terms of feature extraction. If the redundant filters corresponding to some similar feature maps are removed, the recognition accuracy of the original network can still be maintained. Due to the fact that the output of these redundant filters has a certain type of characteristic, it can still be output by retaining similar filters [23].

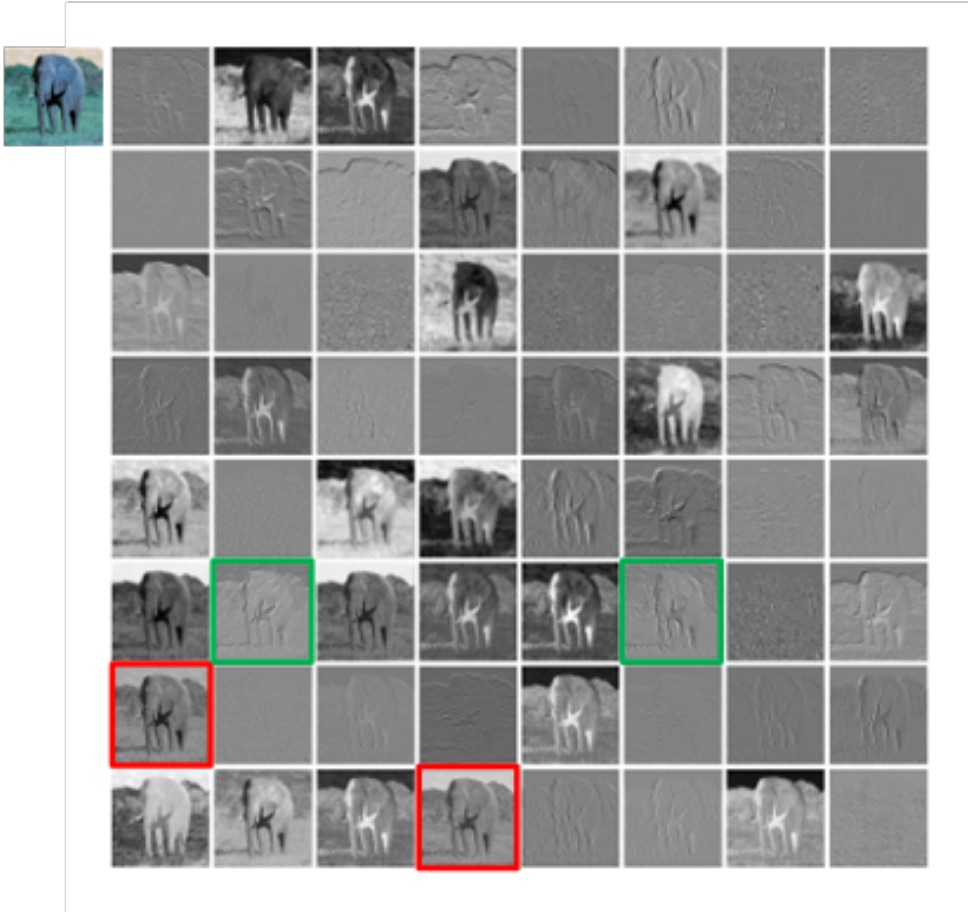

**Figure 1.** Visualization of the feature map of the first convolutional layer of ResNet50 (It can be seen intuitively from the feature map that some feature maps are very similar. As shown in the figure, the visual observation between the two sets of feature maps marked by the red and green boxes is very similar).

### 3.2. Algorithmic Framework

Based on the pruning method of similarity metrics between the filter output feature maps, the pruning process of each convolutional layer is shown in Figure 2 for the pre-trained model to be pruned. It mainly includes three stages: the first stage involves using convolution to calculate the feature map of the current layer; in the second stage, the similarity of each feature map and other feature maps is calculated according to the obtained feature map of the current layer. In the third stage, the similarity score is calculated, and pruning is achieved.

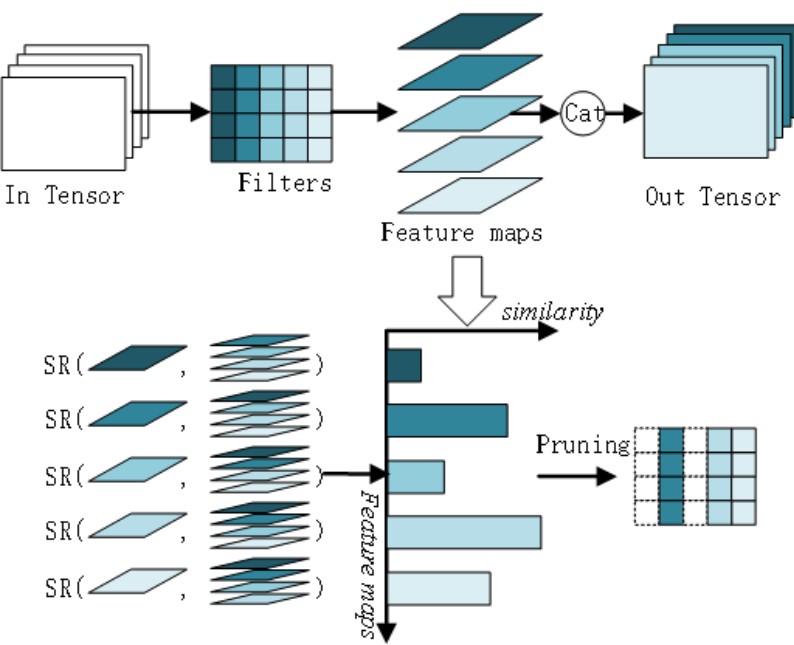

**Figure 2.** Pruning process for each convolutional layer.

The pruning algorithm process is shown in Algorithm 1, and the pruning process includes the following four steps:

(1) Similarity calculation: For a convolutional layer, calculate the similarity between the feature map output of each filter and the feature map output of other filters.
(2) Similarity score calculation: calculate the similarity score for each feature map.
(3) Distance sorting: The similarity score calculated for each feature map is sorted.
(4) Filter and channel pruning: According to the set pruning rate, the feature map with low redundancy is retained to map the corresponding convolution kernels, and the unimportant convolution kernels are removed. Also remove the channel corresponding to the next layer.
(5) Prune the next convolutional layer: Repeat steps 1–4 to prune the next convolutional layer, until all convolutional layers have been pruned.

---

**Algorithm 1** Pruning method based on feature map similarity score.

---

**Require:** pre-trained model, feature map mapping for each layer output, purning rate for each layer $P_i$
**Ensure:** Pruned model
  1: **for** $i$ *in Layers* : **do**
  2:     #Calculate similarity
  3:     **for** $j$ *in features_maps* : **do**
  4:         Calculate the similarity and similarity score SR of the $j$th feature map and all
  5:         other feature maps
  6:     **end for**
  7:     #Pruning
  8:     Sort the filters by the sum of similarities
  9:     Remove unimportant filters
10:     Remove the corresponding channel of layer $i + 1$
11: **end for**
12: #Fine-tuning
13: Fine-tune the pruned model
14: Return pruned model

---

*3.3. Similarity Calculation*

In this paper, the similarity degree is selected to measure the similarity between one feature map $F_{i,j}$ and other feature maps $F_{i,m \neq j}$. To evaluate the performance of evaluating the importance of filters with similarity between feature map maps, multiple feature map similarity indicators are selected for experiments. A total of three indicators were selected: Euclidean Distance (Odist), Difference hashing (dHash), and Structural Similarity (SSIM) to measure the similarity between feature map mappings. These metrics are described below.

(1)    Euclidean distance

Euclidean distance is used to measure the distance between two sample spaces. If the Euclidean distance value is smaller, the two individuals are more similar, and the larger the difference, the greater the difference between the two individuals. An $n$-dimensional Euclidean space is a set of points, each point of which can be represented as $x = (x_1, x_2, \ldots, x_n)$, and the distance between two samples $x$ and $y$ is defined as $Odist(x,y)$. The distance between two points in $n$-dimensional space is defined as shown in the equation.

$$Odist(x,y) = \sqrt{\sum_{i=1}^{n}(x_i - y_i)^2} \tag{1}$$

(2)    Difference hash

The difference hash algorithm is used in similar image retrieval, which can calculate arbitrary pictures and output a fixed-length hash string. Comparing two picture hash strings, the closer the results, the more similar the two pictures are. The difference hash algorithm calculates the hash value through an interpixel gradient. The calculation first converts the image to grayscale and reduces it to $8 \times 8$ pixels. Arranges, in row-first order, the grayscale values of each pixel into a one-dimensional array $I_{8 \times 8}$ that $I_{i,j}$ represents the grayscale values of the pixels in the $i$ row $j$ column. Each pixel is then traversed to calculate its gray difference $d_{i,j}$ from the pixel on the right, which can be calculated using the following formula:

$$d_{i,j} = I_{i,j} - I_{i,j+1} \tag{2}$$

Next, convert these differences to binary values: 0 if the difference is less than or equal to the threshold, otherwise 1, resulting in a binary array $B_{8 \times 7}$ of binary values $B_{i \times j}$ representing the grayscale difference between the pixels in the $i$ row $j$ column and the pixels on the right. Due to the fact that there are only seven differences in a row, each row only needs to be converted to 7 bits of binary value.

Finally, these binary values are combined into a 64-bit hash code $H$ that $H_i$ represents the binary value of the first $i$ bit. It can be calculated using formulas:

$$H = \sum_{i=1}^{64} 2^{i-1} \times B_i \tag{3}$$

where the $B_i$ value of the first $i$ bit in the binary array $B$.

In this way, we obtain a 64-bit hash code, which can be used for image similarity comparison, and the similarity of two images can be calculated by comparing the Hamming distance of their hash codes.

(3)    Structural similarity

When comparing the similarity of two pictures, the brightness, contrast, and structure of the picture are considered at the same time. The higher the value of the structural similarity indicator, the higher the similarity of the two signals. It is calculated as shown in the equation and is a multiplicative combination of three contents.

$$\begin{cases} SSIM(x,y) = [l(x,y)]^{\alpha} \cdot [c(x,y)]^{\beta} \cdot [s(x,y)]^{\gamma}, \\ l(x,y) = \frac{2\mu_x\mu_y + C_1}{\mu_x^2 + \mu_y^2 + C_1}, \\ c(x,y) = \frac{2\sigma_x\sigma_y + C_2}{\sigma_x^2 + \sigma_y^2 + C_2}, \\ s(x,y) = \frac{\sigma_{xy} + C_3}{\sigma_x\sigma_y + C_3} \end{cases} \tag{4}$$

*3.4. Similarity Score*

Use the formula to calculate the sum $i$ $j$ $F_{i,j}$ $F_{i,m}$, $m = 1, 2..M$, $s.t.m \neq j$ of the similarities from the feature map output of the first filter output to all other feature maps.

$$S_j = \sum_{m=1}^{M} similarity(F_{i,j}, F_{i,m}) \tag{5}$$

where $similarity(x, y)$ is the similarity between the sample $x$ and $y$. $m$ indicates the serial number of the first $i$ convolutional layer output feature map.

The larger the value of $S_j$, the more similar the current map is to all the other feature maps. The greater the redundancy of all output feature maps of the current layer, the filter of layer $i$ and the channel of layer $i + 1$ corresponding to this feature map should be removed. After removal, other filters can still extract similar feature maps, which has little impact on the accuracy of the model. Conversely, if the $S_j$ value is smaller, we believe that this feature map is less similar $F_{i,j}$ is to all other feature maps, and the features extracted by the corresponding filter are more important. The corresponding filters and channels should be retained.

In order to prevent individual differences, we take $N$ image samples as input, calculate the sum of Euclidean distances from one feature map to other feature maps multiple times according to the formula, and take the average of the results as a similarity score to measure the feature map $SR$.

$$\begin{aligned} SR_j &= mean(S_j) \\ &= \frac{1}{N} \sum_{1}^{N} \sum_{m=1}^{M} similarity(F_{i,j}, F_{i,m}) \end{aligned} \tag{6}$$

*3.5. Sorting and Pruning*

For the first $i$ convolutional layer, the similarity score of each feature map $\{SR_1, SR_2, \ldots, SR_{N_F}\}$ can be calculated according to the formula. These similarity score results are then sorted in order from largest to smallest to obtain the ordinal index vector of the feature map. Then, combined with the current layer pruning rate $P_i$ and the current layer feature map $N_F$, the number of feature maps to be pruned is determined $N_p = int(P_i \times N_F)$.

Then, the pruning process can be expressed as formula (7) and (8), and a mask matrix is constructed according to the number of feature maps to be pruned $N_P$ and the index vectors sorted by similarity scores. The mask $M$ corresponding to the filter that needs to be pruned is set to 0, and the mask $M$ corresponding to the filter that needs to be retained is set to 1. The filter of the first $i$ layer and the channel of the first $i + 1$ layer is then pruned according to the mask matrix.

$$W' = M \odot W \tag{7}$$

$$C' = M \odot C \tag{8}$$

where $\odot$ is the Hadamard Product (multiplication of the corresponding positional elements of the matrix). Formula (7) represents the process of pruning the weight of the filter, where $M$ is the mask matrix obtained by the method in this paper, and controls the pruning of the filter weight $W$. Formula (8) represents the process of pruning the channel.

In this paper, the pruning of the ResNet network uses a residual connection because the input channel number of the next layer of residual blocks is fixed, so different strategies for pruning the residual structure are different. When pruning, only the first convolutional layer is pruned for the BasicBlock block, and the change in the output dimension of the convolutional layer after pruning will not affect the output dimension of the current residual block. When pruning the second convolutional layer in the residual block, that is, the last convolutional layer, the original output dimension is changed, so the second convolutional layer is not convolved. Similarly, for Bottleneck blocks, only the first two convolutional layers are pruned. The last layer is not pruned, and the dimension of the residual block output is fixed. The pruning position of the residual structure is shown in Figure 3.

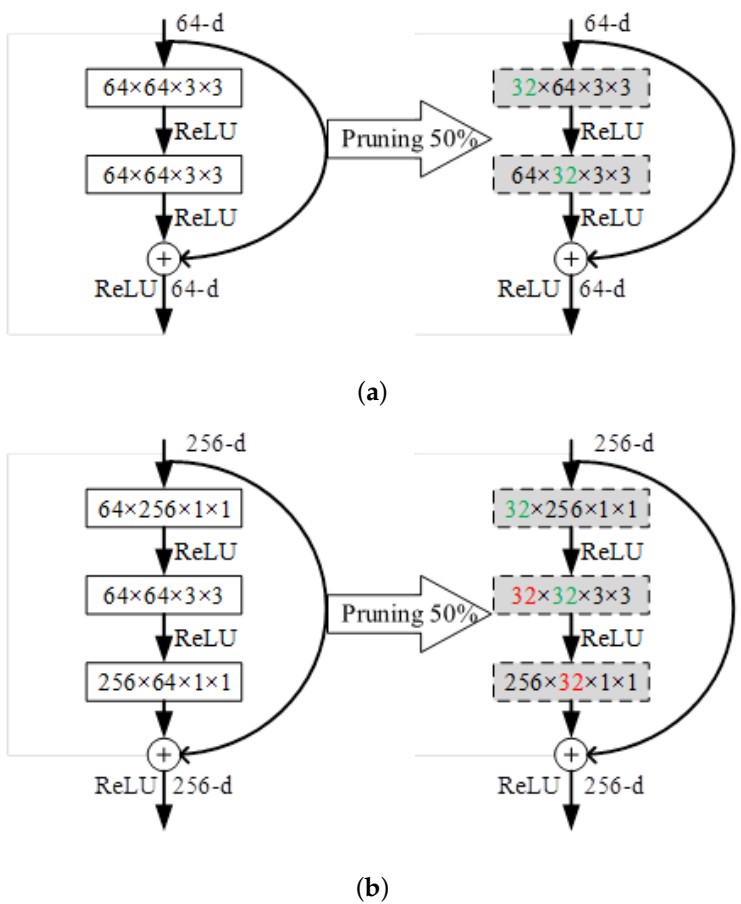

**Figure 3.** Pruning strategy for different residual blocks. (**a**) BasicBlock residual block, with two convolutional layers, prunes only the filter of its first layer. The BasicBlock residual block has two convolutional layers, and only the filter of its first layer is pruned. (**b**) Bottleneck residual block with three convolutional layers pruned only of its first two convolutional layers. The Bottleneck residual block has three convolutional layers, and only the first two convolutional layers are trimmed.

## 4. Experiments

### 4.1. Experimental Environment and Dataset

Experimental environment: The pre training and pruning algorithm implementation of all basic network models in this article are based on the PyTorch framework. The GPU uses A100-SXM4-40 GB of Google Colab Pro platform and 40 GB of graphics memory. The virtual machine RAM size is 81 G. The experimental environment used is Python 3.7, PyTorch version 1.12.1 + cu113. Select SDG as the optimizer, with a batch size of 128, a weight attenuation value of 0.005, and an initial learning rate of 0.01. Randomly select 5 batch sizes with a total of 640 images as samples to calculate similarity redundancy.

Dataset: This article evaluates the method using two datasets. They are the CIFAR-10 dataset and the ISCX-VPN-NonVPN (VPN2016) dataset. CIFAR-10 is a small color image dataset with a total of 100 subcategories. There are 600 images in each category, with sizes of 32 pixels in length and width. 500 images per category are used for training and 100 images are used for validation. The VPN2016 dataset is an encrypted traffic classification dataset that has been processed, with samples uniformly processed to a length of 784 bytes.

*4.2. Evaluation Indicators*

In this paper, three parameters of the model are selected to evaluate the pruning effect. These three indicators are accuracy, parameter amount, and calculation amount, which are explained below.

(1) Accuracy: Accuracy is calculated according to the formula (9). It is used to evaluate the recognition performance of the model. During the pruning process, we are concerned about the change that pruning brings to the accuracy of the model. Use the formula (10) to calculate the change in accuracy of the model after pruning and compression $AccDrop(Acc \downarrow)$.

$$Base\ Acc = \frac{N_{pos}}{N_{all}} \tag{9}$$

where $N_{pos}$ is the number of samples that were correctly identified; $N_{all}$ is the total number of samples.

$$Acc\ Drop(Acc \downarrow) = Acc_{baseline} - Acc_{pruned} \tag{10}$$

where $Acc_{baseline}$ is benchmark model accuracy; $Acc_{pruned}$ is model accuracy after pruning.

(2) Parameters: The number of parameters refers to the number of parameters that the model can train, and it can also be expressed by the size of the storage space occupied by the actual model. Filter pruning is the filter that removes the convolutional layer, and also removes the trainable parameters of the corresponding convolutional layer. The amount of parameters of the $i$th convolutional layer is calculated according to the formula (11).

$$Param = C_F \times (K \times K \times C_I + 1) \tag{11}$$

where $C_F$ represents the number of filters, $K$ represents the size of the filter, and $C_I$ represents the number of the channel.

(3) Floating-point operations (FLOPs): Floating-point operations are the number of floating-point operations required in the model, and the calculation method is a formula (12), which can reflect the complexity of the model. Floating-point operations in the model are mainly addition and multiplication operations. It directly affects the speed of the model and reflects the proportion of model speedup.

$$FLOPs = H \times W \times C_F \times (2 \times K \times K \times C_I) \tag{12}$$

where $H$ and $W$ represent the height and the width of the outputting feature, respectively.

After pruning according to the pruning rate $P$, the number of channels for outputting the feature map becomes $C_F \times (1 - P_i)$, and *FLOPs* correspondingly decreases. The calculation is shown in Equation (13).

$$\begin{aligned} FP_i &= FO_i \times (1 - P_i) \\ &= H \times W \times C_F \times (1 - P_i) \times (2 \times K \times K \times C_I) \end{aligned} \tag{13}$$

If the pruning rate of each layer is specified, the compression rate of *FLOPs*, i.e., the acceleration ratio $R_c$ of the model, can be calculated using the following formula:

$$
\begin{aligned}
R_c &= \frac{\sum_{i=1}^{N_l}(FO_i - FP_i)}{\sum_{i=1}^{N_l} FO_i} \\
&= 1 - \frac{\sum_{i=1}^{N_l} H \times W \times C_F \times (1 - P_i) \times (2 \times K \times K \times C_1)}{\sum_{i=1}^{N_l} H \times W \times C_F \times (2 \times K \times K \times C_I)} \\
&= 1 - \sum_{i=1}^{N_l}(1 - P_i)
\end{aligned}
\tag{14}
$$

In the formula, $N_l$ is the number of convolutional layers.

### 4.3. Similarity Experiment

Firstly, the pruning algorithm in this paper is experimentally verified on the CIFAR-10 dataset, and the two network architectures of ResNet-56 and ResNet-110 are pruned. Pruning uses three measures of similarity: Euclidean distance, difference hash, and structural similarity. Pruning does not use the global pruning rate but sets a different pruning rate for each layer of the network. Set two types of pruning rates: Similarity-Acc and Similarity-Flops: The accuracy priority strategy has a low pruning rate, which can maintain the high accuracy of the network. The compression ratio priority policy pruning rate setting is higher, which can compress more on the network. The pruning rate is determined using the CHIP experiment. A total of 12 sets of experiments were conducted. The pruning results are shown in Table 2.

**Table 2.** Pruning results based on similar redundancy on CIFAR-10 dataset.

| Model | Method | Base Acc (%) | Pruned Acc (%) | Acc ↓ (%) | FLOPs ↓ (%) | Param ↓ (%) |
|---|---|---|---|---|---|---|
| | Odist-Acc | 93.26 | 93.68 | −0.42 | 47.4 | 42.8 |
| | dHash-Acc | 93.26 | 94.20 | −0.94 | 47.4 | 42.8 |
| | SSIM-Acc | 93.26 | 93.86 | −0.60 | 47.4 | 42.8 |
| ResNet-56 | Odist-Flops | 93.26 | 92.58 | +0.68 | 72.3 | 71.8 |
| | dHash-Flops | 93.26 | 92.67 | +0.59 | 72.3 | 71.8 |
| | SSIM-Flops | 93.26 | 92.66 | −0.60 | 72.3 | 71.8 |
| | Odist-Acc | 93.50 | 94.58 | −1.08 | 52.1 | 48.3 |
| | dHash-Acc | 93.50 | 94.53 | −1.03 | 52.1 | 48.3 |
| | SSIM-Acc | 93.50 | 94.35 | −0.85 | 52.1 | 48.3 |
| ResNet-110 | Odist-Flops | 93.50 | 93.29 | +0.21 | 71.6 | 68.3 |
| | dHash-Flops | 93.50 | 93.53 | −0.03 | 71.6 | 68.3 |
| | SSIM-Flops | 93.50 | 93.37 | +0.13 | 71.6 | 68.3 |

ResNet-56 network: Under the strategy of accuracy first, the pruning rate of model FLOPs was 47.4% and the pruning rate of parameters was 42.8%. Based on the three similarity calculation methods, the accuracy of the pruning model was improved. Among them, the pruning method based on a difference hash to measure similarity has the best pruning effect. The accuracy after pruning reached 94.2%, which improved by 0.94% compared with the benchmark model. Under the pruning strategy with compression ratio first, the pruning rate of model FLOPs was 72.3% and the pruning rate of parameters was 71.8%. The accuracy of the model after pruning decreased to 0.68%. After pruning, it has little effect on the accuracy of the model.

ResNet-10 network: Under the strategy of accuracy first, the pruning rate of model FLOPs was 52.1% and the pruning rate of parameters was 48.3%. Based on the three similarity calculation methods, the accuracy of the pruning model was improved. Among them, the pruning method based on Euclidean distance measurement similarity has the best pruning effect. The accuracy after pruning reached 94.58%, which improved by 1.08%

compared with the benchmark model. Under the pruning strategy with compression ratio first, the pruning rate of model FLOPs was 71.6% and the pruning rate of the parameters was 68.3%. The pruning model based on a difference hash to measure similarity still improves the accuracy by 0.21%.

It can be seen from the experimental results that the pruning methods based on similarity scores can restore or even improve the model's performance through model fine-tuning based on pruning compression. The maximum accuracy loss of the three methods is 0.68%. It is proven that the pruning method based on similarity score can effectively remove redundant filters, reduce model size, and maintain network recognition accuracy.

### 4.4. Comparative Experiments

In similarity-based pruning methods, difference hash has achieved relatively good results in both accuracy first and compression first strategies in ResNet-56 and ResNet-110 networks. Among them, pruning based on the similarity of the difference hash method has the best performance. In order to verify the superiority of this method, it is compared with existing methods. The comparison results of evaluation indicators such as decreased accuracy and FLOPS pruning rate are shown in Table 3.

**Table 3.** Comparison of pruning results on CIFAR-10 dataset.

| Model | Method | Base Acc (%) | Pruned Acc (%) | Acc ↓ (%) | FLOPs ↓ (%) | Param↓ (%) |
|---|---|---|---|---|---|---|
| | L1 [14] | 93.04 | 93.06 | −0.02 | 27.6 | 13.7 |
| | HRank [19] | 93.26 | 93.52 | −0.26 | 29.3 | 16.8 |
| | GAL-0.6 [24] | 93.26 | 93.38 | −0.12 | 37.6 | 11.8 |
| | CHIP [20] | 93.26 | 94.16 | −0.90 | 47.4 | 42.8 |
| ResNet-56 | dHash-Acc(Ours) | 93.26 | 94.20 | −0.94 | 47.4 | 42.8 |
| | GAL-0.8 [24] | 93.26 | 91.58 | +1.68 | 60.20 | 65.9 |
| | CHIP [20] | 93.26 | 92.05 | +1.21 | 72.3 | 71.8 |
| | CCEP [25] | 93.48 | 93.72 | −0.24 | 63.42 | - |
| | dHash-Flops(Ours) | 93.26 | 92.67 | +0.59 | 72.3 | 71.8 |
| | GAL-0.5 [24] | 93.50 | 92.74 | 0.76 | 48.5 | 44.8 |
| | CHIP [20] | 93.50 | 94.44 | −0.94 | 52.1 | 48.3 |
| ResNet-110 | dHash-Acc(Ours) | 93.50 | 94.53 | −1.03 | 52.1 | 48.3 |
| | CHIP [20] | 93.50 | 93.63 | −0.13 | 71.6 | 68.3 |
| | CCEP [25] | 93.68 | 93.90 | −0.22 | 67.09 | - |
| | dHash-Flops(Ours) | 93.50 | 93.53 | −0.03 | 71.6 | 68.3 |

For ResNet-56, under the accuracy first strategy, FLOPs can achieve a pruning rate of 47.4%, Param can achieve a pruning rate of 42.8%, and the model accuracy is improved by 0.94%. Compared with the L1, GAL-0.6, HRank, and CHIP methods, the dHash-Acc method achieves the highest accuracy improvement (0.94%) and the highest pruning rate (47.4% on FLOPs and 42.8% on Param). When the pruning rate is higher, the dHash-Flops method still achieves a higher pruning rate (72.3% on FLOPs, 71.8% on Param) compared to GAL-0.8 and CHIP and a lower accuracy decrease (0.59%).

For ResNet-110, FLOPs can achieve a pruning rate of 52.1% under the accuracy priority strategy, while the model accuracy is improved by 1.03%. Compared with the GAL-0.5 and CHIP methods, the dHash-Acc method achieves the highest accuracy (94.53%) and the highest pruning rate (FLOPs reached 52.1% and 48.3% in Param). Compared with CCEP, the accuracy improvement of the dHash-Flops method is only 0.19% lower than that of CCEP, but it has a higher FLOP pruning rate (71.6%).

### 5. Conclusions

In this paper, a pruning method based on feature map similarity score is proposed. We use Euclidean distance, difference hash, and structural similarity to measure similarity between convolutional layer output feature map maps. According to the "similar-

unimportant" principle, redundant filters are removed to reduce the amount of calculation and parameters of the model. Experimental validation was performed on the CIFAR-10 dataset. The three similarity measurement methods can guide filter pruning, and based on model compression, the accuracy of the model can be kept within 1%. The effectiveness of the pruning method based on similar redundancy is proven. Compared with the existing algorithm, the pruning effect has been improved. The proposed pruning method still has room for improvement, such as further improving the compression ratio of the model while ensuring that the accuracy of the model decreases less. Future work will mainly focus on these two aspects.

**Author Contributions:** Conceptualization, J.C. and Z.Y.; methodology, J.C., Z.W. and X.G.; validation, J.C.; writing—original draft preparation, J.C. and X.G.; writing—review and editing, Z.Y. All authors have read and agreed to the published version of the manuscript.

**Funding:** This research received no external funding.

**Data Availability Statement:** The proposed pruning method is experimentally verified on the CIFAR-10 dataset: https://www.cs.toronto.edu/kriz/cifar.html.

**Conflicts of Interest:** The authors declare no conflict of interest.

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
