# Peer review of "A Pruning Method Based on Feature Map Similarity Score"

_2504-2289, doi:10.3390/bdcc7040159_

Round 1

Reviewer 1 Report

This work presents a feature map similarity score-based pruning method. The experimental results show the proposed method can compress the model by more than 70% while maintaining a higher compression ratio and accuracy than traditional methods.

Overall, the research topic is interesting and important. The results are standard and leave room for further clarification. My recommendation is that the manuscript could be accepted for publication with minor revisions. 

Strengths:

+ Paper organization is okay

+ Evaluation part is clear

Major issues:

- The experimental phase should be described in more detail, especially for some important parameters.

- Some limitations can be discussed before the conclusion.

- Writing needs improvements.

See comments to authors

Author Response

Dear Reviewer,
Thank you very much for reviewing our manuscript named “A Pruning method based on feature map similarity score”.
We have taken into consideration your comments and made all the modifications.

General Comments:

Comment 1: The experimental phase should be described in more detail, especially for some important parameters.

Reply: We gratefully appreciate for your valuable comment. Descriptions of experimental parameters and more detailed explanations of important parameters have been added in section 4.1 and section 4.2. Please refer to modified sections in page 8, page 9 and page 10.

Comment 2: Some limitations can be discussed before the conclusion.

Reply: Thank you for your rigorous comment. There is still room for improvement about proposed method, such as further improving the compression ratio of the model while ensuring that the accuracy of the model decreases less. Please refer to line 349 to 351 in page 12.

Comment 3: Writing needs improvements.

Reply: Thank you so much for your careful check. We have carefully checked the entire article and modified some written errors. 

Reviewer 2 Report

In this paper, authors propose a pruning method based on feature map similarity score to remove redundant filters and reduce the amount of calculation and parameters of models. By using Euclidean distance, difference hash, and structural similarity to measure similarity between convolutional layer output feature map maps, filter can be well guided to perform pruning. As the experimental results shown, the effectiveness of the proposed pruning method based on similar redundancy is proven.

But, I have the following questions:

1.     The authors need to explain some important details clearly of “Sorting and pruning” in section 2.5, e.g. “Then the pruning process can be expressed as formulas and formulas” (line 213 in page 7), What are these two formulas and what are the differences?

2.     The author needs to carefully check whether the citations are correct, e.g. “Because the output of these redundant filters has a certain type of characteristic, it can still be output by retaining similar filters [? ].” (line 130 in page 3).

3.     Some descriptions of the figures are inconsistent with them, e.g. Figure 3(a) and Figure 3(b).

4.     The authors need to clearly describe the meaning of parameters in the formulas, e,g formula (11) and formula (12).

5.     It is suggested to check the spelling and grammar issues throughout the paper, and some obvious writing mistakes should be corrected, e.g. Figuree -> Figure (line 133 in page 4).

There are some typos in this paper, which should be carefully checked.

Author Response

Dear Reviewer,
Thank you very much for reviewing our manuscript named “A Pruning method based on feature map similarity score”.
We have taken into consideration your comments and made all the modifications.

Comment 1: The authors need to explain some important details clearly of “Sorting and pruning” in section 2.5, e.g. “Then the pruning process can be expressed as formulas and formulas” (line 213 in page 7), what are these two formulas and what are the differences?

Reply: We gratefully appreciate for your valuable comment. Formula (7) represents the process of pruning the weight of the filter, where M is the mask matrix obtained by the method in this paper, and controls the pruning of the filter weight W. Formula (8) represents the process of pruning the channel.” This clearer description has been added to line 227 on page 7.

Comment 2: The author needs to carefully check whether the citations are correct, e.g. “Because the output of these redundant filters has a certain type of characteristic, it can still be output by retaining similar filters [? ].” (line 130 in page 3).

Reply: Thank you so much for your careful check. This issue has been corrected on page 4, line 136, and the entire text has been checked for writing issues. 

Comment 3: Some descriptions of the figures are inconsistent with them, e.g. Figure 3(a) and Figure 3(b).

Reply: Thank you so much for your careful check. We have made changes to the descriptions of Figure 3 (a) and Figure3 (b), which are located on page 8, line 239.

Comment 4:  The authors need to clearly describe the meaning of parameters in the formulas, e,g formula (11) and formula (12).

Reply: We gratefully appreciate for your valuable comment. We have added the following description to formula (11) on page 8, line 272: where CF represents the number of filters, K represents the size of the filter and CI represents the number of the channel. We have added the following description to formula (12) on page 8, line 279: where H and W represent the height and the width of the outputting feature, respectively. We have added clearer descriptions to other formula pages in the article.

Comment 5:  It is suggested to check the spelling and grammar issues throughout the paper, and some obvious writing mistakes should be corrected, e.g. Figuree -> Figure (line 133 in page 4).

Reply: Thank you so much for your careful check. This issue has been corrected at line 139 on page 4, and we have corrected 'figuree' to 'figure'. Further comprehensive checks have been conducted on the grammar and spelling, and modifications have been made to the identified issues and other issues.

Reviewer 3 Report

What is the time complexity of this algorithm, regarding that a lot of distance calculation is required? 

What bothers me is the lack of statistics of results. Results are given as fixed values, while multiple experiments really give different values, so there is a mean and some std. deviation. Even if these results are mean values, it is a question whether two results are statistically significantly different, since they may have large stdev. The t-test or ANOVA can reveal that. 

Throughout the paper, there is a plenty of missing spaces, usually close to brackets, such as theory[1-4], He[18]et al., etc. This is not acceptable

line 33: 528 million of what?

line 37: compression acceleration ?

lines 102-105: sentences are not correct - is this imperative speech?

line 116: This chapter prunes ?

130: ?

150: , until 

map map 

158: European distance?!

190 Similar score? probably Similarity score - the same thing in many places in the paper

193: where should not be indented (as well as some other occurences)

205: sentence?

213: as formulas and formulas ?

Mainly acceptable, except in some cases (see Suggestions)

Author Response

Dear Reviewer,
Thank you very much for reviewing our manuscript named “A Pruning method based on feature map similarity score”.
We have taken into consideration your comments and made all the modifications.

Comment 1: What is the time complexity of this algorithm, regarding that a lot of distance calculation is required? 

Reply: In fact, the pruning method proposed in this paper is based on similarity calculation (Odist, dHash or SSIM), so as to compress the deep learning models. After pruning the deep learning model, the number of parameters and computational complexity of the model will decrease. Therefore, using different similarity calculation methods will not increase the complexity of the model.

Comment 2: What bothers me is the lack of statistics of results. Results are given as fixed values, while multiple experiments really give different values, so there is a mean and some std. deviation. Even if these results are mean values, it is a question whether two results are statistically significantly different, since they may have large stdev. The t-test or ANOVA can reveal that. 

Reply: Table 2 and Table 3 show the experimental results of different algorithms pruning the model on the same dataset. From the table, it can be seen that different algorithms have different compression effects on the model, and their impacts on the accuracy of the model are also different. Meanwhile, the experimental results do not reflect multiple experimental results of a certain algorithm on the same dataset. We just want to reflect the performance of different pruning algorithms through these two tables.

Comment 3: Throughout the paper, there is a plenty of missing spaces, usually close to brackets, such as theory[1-4], He[18]et al., etc. This is not acceptable.

Reply: The above two questions have been corrected on line 24 of the page 1 and line 96 of the third page, respectively. Further comprehensive checks have been conducted on the grammar and spelling of the entire text, and modifications have been made to the identified issues and other issues.

Comment 4: line 33: 528 million of what?

Reply: This issue has been corrected on line 33 of the page 1 by changing “528 million” to “528 MB”.

Comment 5: line 37: compression acceleration ?

Reply: We have replaced the original text at line 37 on page 2 with “Therefore, it is necessary to study how to compress the model in order to improve its efficiency”.

Comment 5: lines 102-105: sentences are not correct - is this imperative speech?

Reply: We have made the following corrections to the original text: There is another type pruning method based on similarity analysis, considering the similarity between the output feature maps of convolutional layer. This type of method clusters the features output from the convolutional layer, preserves representative feature maps, and then removes filters and channels corresponding to redundant feature maps. However, this type of method often fails to select an appropriate number of cluster centers. Please refer to line 113 to 117 in page 4. We have checked the writing issues throughout the entire article.

Comment 6: line 116: This chapter prunes ?

Reply: We have corrected 'This chapter prunes based on the similarity between the feature maps of each filter output.' to 'In this paper, the proposed pruning algorithm is based on similarity calculation between the feature maps of each filter output.' Please refer to line 122 in page 4.

Comment 7:  ?

Reply: We have corrected this issue by adding '?' changed to '[23]' (line 136 in page 4), and checked for writing issues throughout the entire text.

Comment 8:  , until 

Reply: We have corrected 'Repeat steps 1-4 to prune the next convolutional layer. Until all convolutional layers have been pruned.' to 'Repeat steps 1-4 to prune the next convolutional layer, until all convolutional layers have been pruned'. Please refer to line 155 in page 5.

Comment 9: map map 

Reply: We have corrected ‘feature map map’ to ‘feature map mapping’. Please refer to Algorithm 1 in page 5.

Comment 10: European distance?!

Reply: ‘European distance’ has been modified to ‘Euclidean distance’. Please refer to line 164 om page 6. Further comprehensive checks have been conducted on the grammar and spelling of the entire text, and modifications have been made to the identified issues and other issues.

Comment 11: where should not be indented (as well as some other occurences).

Reply: We have corrected this issue in line 200. We have checked the writing issues of the entire text.

Comment 12: sentence?

Reply: We have corrected this issue in line 212, and checked the grammar of the entire text.

Comment 13: as formulas and formulas ?  

Reply: Formula (7) represents the process of pruning the weight of the filter, where M is the mask matrix obtained by the method in this paper, and controls the pruning of the filter weight W. Formula (8) represents the process of pruning the channel. This clearer description has been added to line 227 on page 7.

Reviewer 4 Report

The paper presents a hashing-based filtering mechanism on feature maps in for CNN-based methods. The method's effectiveness is demonstrated on ResNet-56 and ResNet-110 on the CIFAR-10 dataset.

Suggestions for improvement.

(1) Related works should be presented as a separate section (rather than as a part of the introduction).

(2) A summary of related works should be provided as a table.

(3) The presentation of Table 2 is confusing. It should be modified.

(4) Some ablation studies on important hyperparameters could be included.

(5) European distance  --> Euclidean distance

Author Response

Dear Reviewer,
Thank you very much for reviewing our manuscript named “A Pruning method based on feature map similarity score”.
We have taken into consideration your comments and made all the modifications.

Comment 1: Related works should be presented as a separate section (rather than as a part of the introduction).

Reply: We have made modifications to the structure of the article and added a section titled "Related Works" at line 53 on page 2. 

Comment 2: A summary of related works should be provided as a table. 

Reply: Our summary of the related works has been summarized into a table and added to page 3, line 109.

Comment 3: The presentation of Table 2 is confusing. It should be modified.

Reply: The previous name of Table 2 has been changed to Table 3.We have corrected the description in Table 3 and added a more detailed description. The specific modifications are located on page 11, lines 327 to 339.

Comment 4:  Some ablation studies on important hyperparameters could be included.

Reply: This part of the experiment can be supplemented, but we believe that this article only introduces similarity for pruning, so there is no need for ablation experiments

Comment 5: European distance  --> Euclidean distance

Reply: Change to ‘Euclidean distance’ .Located on page 6, line 164.Further comprehensive checks have been conducted on the grammar and spelling of the entire text, and modifications have been made to the identified issues and other issues.